# DUAL GRAINED QUANTIZATION: EFFICIENT FINE-GRAINED QUANTIZATION FOR LLM

## ABSTRACT

Large Language Models (LLMs) pose significant hardware challenges related to memory requirements and computational ability. There are two mainstream quantization schemes for LLMs: coarse-grained (*e.g.,* channel-wise) quantization and fine-grained ( *e.g.,* group-wise) quantization. Fine-grained quantization has smaller quantization loss, consequently achieving superior performance. However, when applied to weight-activation quantization, it disrupts continuous integer matrix multiplication, leading to inefficient inference. In this paper, we introduce Dual Grained Quantization (DGQ), a novel A8W4 quantization for LLM that maintains superior performance while ensuring fast inference speed. DSQ dequantizes the fine-grained INT4 weight into coarse-grained INT8 representation and preform matrix multiplication using INT8 kernels. Besides, we develop a two-phase grid search algorithm to simplify the determination of fine-grained and coarse-grained quantization scales. We also devise a percentile clipping schema for smoothing the activation outliers without the need for complex optimization techniques. Experimental results demonstrate that DGQ consistently outperforms prior methods across various LLM architectures and a wide range of tasks. Remarkably, by our implemented efficient CUTLASS kernel, we achieve **1.12** $\times$ memory reduction and **3.24** $\times$ speed gains comparing A16W4 implementation. These advancements enable efficient deployment of A8W4 LLMs for real-world applications.

## 1 INTRODUCTION

The internet has generated vast amounts of text data, and with the increase in computing power, Large Language Models (LLMs) such as GPT-4 (Bubeck et al., 2023) have excelled in comprehending and generating natural language. However, these models have become much larger. For instance, GPT-3 (Brown et al., 2020) boasts over 175 billion parameters. Open-source models like OPT (Zhang et al., 2022) and BLOOM (Scao et al., 2022), built on the GPT architecture, often surpass GPT-3 in parameter count. More recently, models like Meta's 65B (Touvron et al., 2023a) have matched GPT-3's language generation abilities. To put this in perspective, LLaMA-65B is approximately 190 times larger than BERT-Large (Devlin et al., 2018), necessitating around 130 GB of memory storage, which requires two A100 GPUs to accommodate.

Model quantization, as discussed in Han et al. (2015), maps high-precision values to lower-precision representations (*e.g.,* INT8, INT4, FP8, FP4). This technique serves to reduce memory requirements and enhance inference speed. In the context of Large Language Models (LLM), Post-training Quantization(PTQ) (Nagel et al., 2020; Hubara et al., 2020) methods are preferred, primarily due to the extensive computational demands associated with fine-tuning for Quantization Aware Training (QAT) (Esser et al., 2019; Martinez et al., 2018). To maintain precision when applying PTQ, we incorporate finer-grained quantization methods (Yao et al., 2022; Bondarenko et al., 2021). Fine-grained quantization involves dividing a dimension into multiple parts and quantizing each smaller slice, while coarse-grained quantization (Xiao et al., 2023; Yuan et al., 2023) typically quantizes the entire tensor or quantizes along a dimension. Fine-grained quantization is commonly used in weight-only quantization (Frantar et al., 2022; Lin et al., 2023; Dettmers et al., 2023b), where only model weights are quantized. However, in weight-activation quantization (Xiao et al., 2023; Yuan et al., 2023; Wei et al., 2023), which includes both weights and activations, a key challenge emerges due to varying quantization scales along the accumulation axis, typically represented by input channels in linear layers.

Addressing this challenge involves partitioning the integer General Matrix Multiplication (GEMM) operation into discrete segments and aggregating them through floating-point arithmetic. Notably, implementing such a scheme on existing hardware infrastructure proves to be notably inefficient.

To address this challenge, we introduce Dual Grained Quantization (DGQ) as an efficient deployment solution for LLMs. DGQ combines the performance benefits of fine-grained quantization with the computational efficiency of coarse-grained methods. In our weight quantization approach, we employ a two-step process. To avoid the need for segmenting General Matrix Multiplication (GEMM) operations, we dequantize INT4-weight back to INT8, instead of directly casting to INT8. This results in INT8 weights with a coarser-grained scale. As a consequence, our coarser-grained INT8 activation-weight quantization can be efficiently

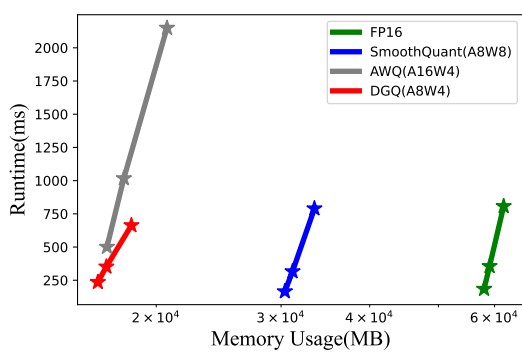

**Figure 1. Comparison of four different quantization methods in terms of runtime and memory usage.** Opt-30b with different sequence length from 512 to 2048 is used as the baseline. Our A8W4 implement maintain the comparable run time to A8W8 and FP16 while maintaining small memory usages. Besieds, our implement has a smaller memory usage than A16W4 implement.

accelerated by general-purpose hardware, eliminating the necessity for specialized hardware designs. DGQ introduces two quantized scales for weight tensors: a fine-grained INT8 quantization scale and a coarse-grained FP16 quantization scale. A notable challenge arises when directly quantizing the FP16 fine-grained scale, leading to a significant drop in accuracy, as illustrated in Table (1). To tackle this issue, we propose a two-phase search approach. In the first phase, we diligently search for the original fine-grained parameters. In the subsequent phase, we determine coarse-grained parameters using the original weights, rather than relying on previously derived fine-grained parameters. We then partition the fine-grained parameters using the determined coarse-grained parameters, preserving the performance of the original fine-grained quantized models.

Recent studies (Xiao et al., 2023; Yuan et al., 2023; Wei et al., 2023) have made significant strides in addressing the challenge of outliers in Large Language Models (LLMs). These efforts have resulted in INT8 models that match the accuracy levels of their full-precision counterparts. Drawing inspiration from LLM.int8() (Dettmers et al., 2022) and AWQ (Lin et al., 2023), we propose a novel percentile clipping smoothing strategy to further enhance quantization. One of the notable advantages of our method is its gradient-free nature, which ensures the preservation of the model's generalization ability across diverse domains and modalities. Our optimization approach also demonstrates exceptional efficiency. For instance, we can achieve quantized A8W4 models for BLOOM-176B within just one GPU hour on the A100-80G.

The experimental results highlight the effectiveness of DGQ across a wide range of tasks, model families, and sizes. In particular, DGQ demonstrates impressive performance on the WikiText-2 (Merity et al., 2016) task, achieving a mere 0.3 perplexity loss for integer A8W4 models. This outperforms the floating-point A8W4 quantization scheme (Wu et al., 2023) by approximately 0.3. To support DGQ inference, we have implemented efficient CUTLASS kernels. Leveraging these efficient kernels, DGQ achieves a remarkable up to 3x speedup compared to the A16W4 baseline while maintaining similar memory usage. Furthermore, DGQ excels in overhead management, enabling it to achieve similar inference speeds to A8W8 models but with only half the memory usage.

## 2 RELATED WORKS

### 2.1 LARGE LANGUAGE MODELS

Pre-trained large language models (LLMs), such as GPT-4 (Bubeck et al., 2023), LLaMA-2 (Touvron et al., 2023b), and OPT (Zhang et al., 2022), have demonstrated exceptional performance across a wide range of tasks and domains. Despite their impressive capabilities, these models come with significant memory and computational demands, presenting challenges in practical deployment. To address these challenges, a growing body of research has emerged, focusing on various techniques

to optimize LLMs. These approaches encompass model compression methods (Frantar & Alistarh, 2023; Xiao et al., 2023), distributed computing strategies (Aminabadi et al., 2022), and computational graph optimizations (Dao et al., 2022; Dao, 2023). In this study, our primary focus is on model quantization, a key component of model compression. We aim to explore and advance quantization techniques to enable efficient deployment of LLMs.

## 2.2 MODEL QUANTIZATION

Quantization-Aware Training (QAT) (Esser et al., 2019; Martinez et al., 2018) and Post-training Quantization (PTQ) (Choukroun et al., 2019; Li et al., 2021; Wei et al., 2022). QAT fine-tunes quantized models with the full dataset, preserving accuracy but involving complex computations, making it less suitable for LLMs. In contrast, PTQ directly quantizes models with little data and computation. Techniques like AdaRound (Nagel et al., 2020), and Adaquant (Hubara et al., 2020) optimize quantization parameters and distill quantized models layer by layer. Some approaches (Qin et al., 2022; He et al., 2022) employ alternating optimization, and some push the boundaries by compressing transformations into binary values. These quantization techniques are crucial for enhancing the efficiency of deep learning models, especially in resource-constrained environments.

For LLMs quantization, fine-grained quantization is introduced to solve the significant accuracy drop. PEG-PTQ (Bondarenko et al., 2021) proposed per-embedding-group quantization, which splits the activation into several groups and quantizes activation via each group. ZeroQuant (Yao et al., 2022) used token-wise activation quantization and group-wise weight quantization as quantization scheme. LLM.int8() (Dettmers et al., 2022) finds that the outliers in activation have a significant contributor to poor quantization performance and splits the outliers and calculate the matrix multiple evolved outliers in FP16. GPTQ (Frantar et al., 2022) and SparseGPT (Frantar & Alistarh, 2023) use second-order approximation to quantize and prune weights. GPTQ also introduces channel-reorder via hessian matrix to weight to maintain better accuracy. RPTQ (Yuan et al., 2023) reorders channels and splits activation into three groups via activation ranges. It successfully quants LLMs into fine-grained A4W4 models.SmoothQuant (Xiao et al., 2023) proposes a mathematically equivalent per-channel scaling transformation makes activation easier to quantize. Outlier Suppression+ (Wei et al., 2023) introduce a fusible offset for activation quantization and convert the hard-to-quant asymmetric quantization into symmetric quantization. AWQ (Lin et al., 2023) finds that the sensitive weight is based on significant activation channels and introduces a scale to amplify the significant weight channels. OmniQuant (Shao et al., 2023) follows the Quadapter (Park et al., 2022) try to only optimize the smooth scale during optimization.

ZeroQuantv2 (Yao et al., 2023) introduces the Low Rank Compensation (LoRC) technique to enhance the precision of A8W4 quantized models, resulting in improved accuracy. However, it's essential to note that LoRC involves FP16 computations, which add computational overhead. ZeroQuant-FP (Wu et al., 2023) explores the use of A8W4 precision within a floating-point (FP) context, showcasing improved accuracy. It introduces an efficient method for optimizing weight scales, especially relevant given the limited support for FP8, primarily on H100 hardware. However, it's worth noting that floating-point calculations generally entail higher computational costs compared to integer-based computations. QLoRA (Dettmers et al., 2023a) introduces the NF4 double quantization technique, which enhances memory utilization by quantizing FP32 group-wise quantization scales to FP8 while using a channel-wise FP32 scale. This method prioritizes memory efficiency over computational efficiency and shows that direct quantization of quantization scales to FP8 does not lead to substantial accuracy loss. In our proposed methods, we tackle the dual quantization challenge for integer (INT) values by introducing a lossless compression solution. This innovative approach seamlessly combines computational efficiency with memory conservation, presenting a promising solution within this field.

## 3 METHOD

### 3.1 PRELIMINARY

Quantization is a crucial process that transforms high-precision values into low-precision representations. In our work, we emphasize the use of uniform integer quantization, which offers better hardware support and computational efficiency. Asymmetric quantization can be expressed as follows:

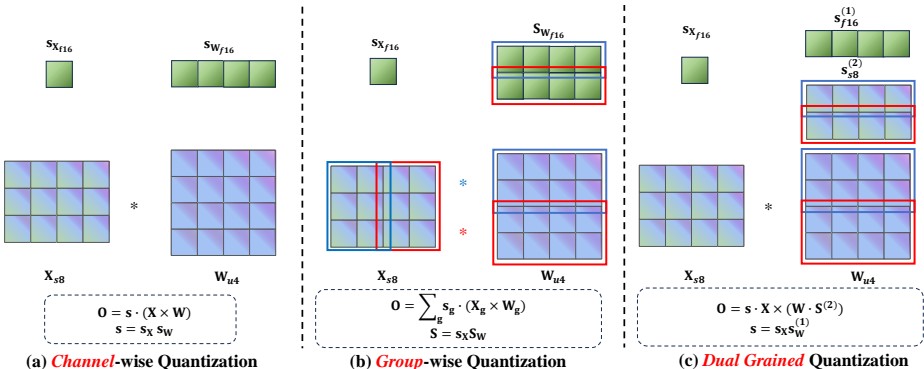

**Figure 2. Different grained quantization method.** Unlike channel-wise and DGQ quantization, group-wise quantization tends to result in FP16 accumulation, which is conflict with typical hardware design.

$$\widehat{\mathbf{X}} = \mathcal{Q}(X, \mathbf{s}, \mathbf{ZP}) = (\mathbf{clamp}(\left\lfloor \frac{\mathbf{X}}{\mathbf{s}} \right\rceil + \mathbf{ZP}, 0, 2^N - 1) - \mathbf{ZP})\mathbf{s}, \quad (1)$$

$$\mathbf{ZP} = \left\lfloor \frac{\alpha \mathbf{min}(\mathbf{X})}{\mathbf{s}} \right\rceil, \quad \mathbf{s} = \frac{\alpha(\mathbf{max}(\mathbf{X}) - \mathbf{min}(\mathbf{X}))}{2^N - 1}$$

Here, $\mathbf{X}$ represents the full-precision tensor, $\widehat{\mathbf{X}}$ is the quantized counterpart, $\mathbf{s}$ denotes the quantization scale, and $N$ is the number of bits. For symmetric cases, the zero-point $\mathbf{ZP}$ is set to zero, and the quantization scale $\mathbf{s}$ is computed as $\mathbf{max}(|\mathbf{X}|)/(\mathbf{2^{N-1}} - \mathbf{1})$.

There are two distinct quantization methods for LLM quantization, weight-only quantization and weight-activation quantization. Weight-only quantization focuses on quantizing only the model weights into low-bit representations while preserving full precision during inference. This method reduces memory requirements and storage demands. In contrast, weight-activation quantization extends quantization to both the model weights and input activations. By utilizing lower-bit representations for both weights and activations, this approach accelerates inference. In our paper, we primarily concentrate on weight-activation quantization, exploring its advantages and optimizing it to enhance overall model efficiency.

## 3.2 DUAL GRAINED QUANTIZATION

For weight quantization, we consider two levels of granularity, *i.e.* channel- and group-wise quantization, as illustrated in Figure 2 (a) and (b). Channel-wise quantization involves assigning a specific quantization scale to each output channel, which is then applied to the result of the integer General Matrix Multiplication (GEMM) operation. On the other hand, group-wise quantization is a more refined approach that divides the output channels into multiple groups and assigns a quantization scale to each group. These quantization scales are collectively represented as a matrix denoted by $\mathbf{S}$. In general, group-wise quantization tends to yield smaller quantization errors compared to channel-wise quantization, resulting in higher accuracy. To achieve model compression with lower bit precision, it becomes essential to introduce fine-grained quantization (*e.g.,* group-wise) methods into weight quantization. However, group-wise quantization methods present a notable challenge when it comes to their implementation on hardware platforms. In group-wise quantization, the reduction axis is divided into multiple groups, each with distinct quantization scales. These groups effectively operate within separate INT8 domains, which poses limitations on their ability to perform direct INT8 General Matrix Multiplication (GEMM) operations. To facilitate Group-Wise INT8 quantization, the matrix is divided into segments, allowing for separate GEMM computations on each segment. Following these computations, the results are combined through FP16 accumulation. It's worth noting, however, that the adoption of FP16 accumulation introduces a delay in the INT8 kernel's GEMM calculations.

To enhance the hardware efficiency of group-wise quantization, we introduce dual-grained quantization (DGQ) as shown in Figure 2 (c). This approach incorporates a crucial dequantization step before the GEMM operation, which involves converting INT4 weights into INT8 weights, enabling subsequent INT8 GEMM operations. Our method leverages group-wise INT8 quantization scales and

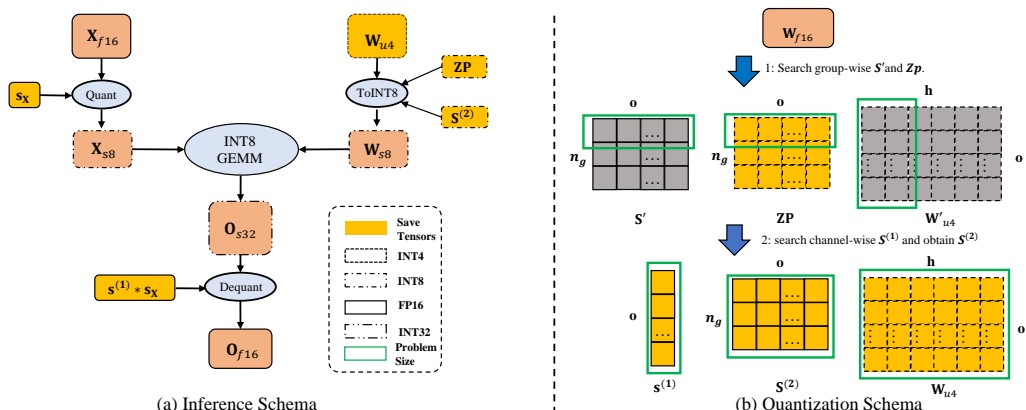

**Figure 3. DGQ inference schema and quantization schema with Two-Phase Search algorithm.**

zero points to transform INT4 weights into INT8 weights. Following this, we employ channel-wise FP16 quantization scales to determine the dequantization factors for the output. Given the hidden states $\mathbf{X}_{s8} \in \mathbb{N}_{s8}^{b \times h}$, weight matrix $\mathbf{W}_{u4} \in \mathbb{N}_{s8}^{h \times o}$, the quantization process is defined by the following equations:

$$
\begin{aligned}
\mathbf{s}_{f16} &= \mathbf{s}_{\mathbf{X}\,f16} \cdot \mathbf{s}^{(\mathbf{1})}{}_{f16}, \\
\mathbf{W}_{s8} &= \mathbf{S}^{(\mathbf{2})}{}_{s8} \cdot (\mathbf{W}_{u4} + \mathbf{ZP}_{u4}), \\
\mathbf{O}_{f16} &= \mathbf{s}_{f16} \cdot (\mathbf{X}_{s8}\mathbf{W}_{s8}),
\end{aligned}
\tag{2}
$$

Here, $\mathbf{S}_{s8}^{(2)} \in \mathbb{N}_{s8}^{g \times o}$ represents the group-wise quantization scales, $\mathbf{ZP}_{s8} \in \mathbb{N}_{s8}^{g \times o}$ represents the group-wise quantization zero points and $\mathbf{s}^{(\mathbf{1})}{}_{f16} \in \mathbb{R}^{o}$ represents channel-wise quantization scales. $o$ denotes the number of output channels for the weight, $\mathbf{n_g}$ signifies the group number of group-wise quantization and $\mathbf{g}$ denotes the group size of group-wise quantization, such that $\mathbf{n_g} \cdot \mathbf{g} = \mathbf{h}$, where $h$ represents the input channels for the weight. When comparing A16W4 to A8W4 with DGQ, it becomes evident that the latter configuration significantly accelerates the calculation process. Moreover, in comparison to A8W8, A8W4 with DGQ exhibits reduced weight memory usage and facilitates faster memory transfers. Our method offers a significant improvement in calculation matrix efficiency when compared to weight-only quantization with the same weight bitwidth. Furthermore, it demonstrates superior memory efficiency compared to weight-activation quantization with lower bit-width weights.

### 3.3 TWO-PHASE GRID SEARCH

A significant quantization error is introduced when directly quantizing group-wise parameters $\mathbf{S}'$ into our quantization schema $\mathbf{s}^{(\mathbf{1})}$ and $\mathbf{S}^{(\mathbf{2})}$, as depicted in Table 1. Furthermore, searching for three key parameters directly results in an excessively large search space. To provide some context, let $\sigma_o$ represent the grid size for $\mathbf{s}^{(\mathbf{1})}$ and $\sigma_g$ represent the grid size for $\mathbf{S}^{(\mathbf{2})}$. In the case of dual-grained quantization (DGQ), the size of the search space becomes substantial, with a complexity of $\mathbf{O}(\mathbf{o} \times \mathbf{n_g} \times \sigma_{\mathbf{o}} \times \sigma_{\mathbf{g}})$. While it is possible to parallelize the search algorithm along the $\mathbf{o}$ dimension, the sheer magnitude of the search space remains a formidable challenge, scaling as $\mathbf{O}(\mathbf{n_g} \times \sigma_{\mathbf{o}} \times \sigma_{\mathbf{g}})$. To address these two problem, we propose a novel two-phase search approach.

**Table 1. WikiText-2 Results for A8W4 LLaMA family: Original Group-Wise Quantization vs. Round to Nearst(RTN) Dual-Grained Quantization.**

| LLaMA PPL $\downarrow$ | 1-7B | 1-13B | 1-30B | 2-7B | 2-13B |
|---|---|---|---|---|---|
| Original | 6.03 | 5.39 | 4.43 | 5.87 | 5.23 |
| RTN | 6.93 | 6.00 | 4.83 | 2155.54 | 5.48 |

To narrow down the expansive search space, our initial step involves quantizing the weights into group-wise INT4 representations. This approach capitalizes on the assumption of weight independence across in-channels. We divide both weights and activations into $\mathbf{n_g}$ segments, based on the group size $g$ along the channel dimension ($h$), treating each segment as an independent matrix multiplication operation. To enhance parallelism, we introduce the parameter $o$ as the parallelism axis, and our goal is to formulate the minimum granularity optimization problem as follows:

$$\underset{\mathbf{s}'_k, \mathbf{zp}_k}{\arg\min} \|\mathbf{X}_{[:,(k-1)g:kg]}\mathbf{W}_{[(k-1)g:kg,:]} - \widehat{\mathbf{X}}_{[:,(k-1)g:kg]}\mathcal{Q}_W(\mathbf{W}_{[(k-1)g:kg,:]}, \mathbf{s}'_k, \mathbf{zp}_k)\|^2. \quad (3)$$

Here, $k$ denotes the $k$-th group with group size $g$, and $\mathbf{s}'_k$, $\mathbf{zp}_k$ represent $k$-th row of quantization scale $\mathbf{S}'$ and zero point $\mathbf{ZP}$. $\widehat{\mathbf{X}}$ signifies the dequantized activation. We employ the grid search for the scaling parameter $\alpha$ (Eq. 1) to determine the optimal $\mathbf{s}'_k$, $\mathbf{zp}_k$. The time complexity is $\mathbf{O}(n_g \times \sigma_g)$.

For next phase, we aim to decouple the group-wise FP16 scale $\mathbf{S}'$ into two distinct components: a channel-wise FP16 scale $\mathbf{s^{(1)}}$ and a group-wise INT8 scale $\mathbf{S^{(2)}}$. To avoid the overflow or underflow for integer representation, we impose the following constraints for INT8 data $\mathbf{S^{(2)}}$, $\mathbf{W}_{s8}$ and $\mathbf{W}_{u4}$.

$$\mathbf{W}_{u4} \in [\mathbf{0}, \mathbf{15}], \mathbf{S^{(2)}} \in [-\mathbf{128}, \mathbf{127}], \mathbf{W}_{s8} \in [-\mathbf{128}, \mathbf{127}] \quad (4)$$

Remarkably, we can consolidate the range for $\mathbf{W_{s8}}$ into the range of $\mathbf{W_{u4}}$. The proof of this consolidation can be found in Appendix B. This fusion leads us to the following constraint for our search space:

$$\mathbf{W}_{u4} \in [\mathbf{max}(\mathbf{0}, \lfloor\frac{-\mathbf{127}}{\mathbf{S^{(2)}}}\rceil + \mathbf{ZP}), \mathbf{min}(\mathbf{15}, \lfloor\frac{\mathbf{127}}{\mathbf{S^{(2)}}}\rceil + \mathbf{ZP})] \quad (5)$$

Similar to the prior phase, we also use the grid employing for the scaling parameter $\alpha$ (Eq. 1) and derive the channel-wise parameter $\mathbf{s^{(1)}}$. The optimization problem for this phase is as follows:

$$\underset{\mathbf{s^{(1)}}}{\arg\min} \|\mathbf{XW} - \widehat{\mathbf{X}}\mathcal{Q}_W(\mathbf{W}, \mathbf{S}, \mathbf{ZP})\|^2, \mathbf{S^{(2)}} = \lfloor\frac{\mathbf{S}'}{\mathbf{s^{(1)}}}\rceil, \mathbf{S} = \mathbf{s^{(1)}} \cdot \mathbf{S^{(2)}} \quad (6)$$

Here, we replace the max value and min value in Eq. 1 with Eq. 5 to prevent overflow or underflow in integer representations. It's worth noting that this phase allows for efficient parallelization along the $\mathbf{o}$ dimension and exhibits a time complexity of $\mathbf{O}(\sigma_\mathbf{c})$. As a result, the total time complexity for our approach becomes $\mathbf{O}(\mathbf{n_g} \times \sigma_\mathbf{g} + \sigma_\mathbf{c})$. This approach significantly reduces the search space compared to the one-step search. Experimental results demonstrate that our method introduces virtually no additional error.

## 3.4 PERCENTILE CLIPPING SMOOTHING

As discussed in prior works (Xiao et al., 2023), outliers in activation have a large impact on model accuracy. A channel-wise smooth would help limit the influence of outliers for activation quantization. For weight quantization, AWQ (Lin et al., 2023) proposed to use activation outliers to amplify the weight in outlier channels to decrease the quantization error of such channels. This is similar in principle to reordering in GPTQ (Frantar et al., 2022). As the quantization-sensitivity can be approximated by the Hessian matrix of weights. Hessian matrix of weights is calculated by $\mathbf{H} = \mathbf{XX^T}$ and is directly related to the absolute value of the input. Therefore, the mean value of the input channel for activation can approximately present the quantization sensitivity of weight.

For weight-activation smooth scales, the quantization difficulty that moves from activation to weight could act as the quantization amplifier for weights, making it possible as a win-win for both weight and activation quantization. And from the observation from Dettmers et al. (2023b), the outliers always appear at fixed channels, which means that the biggest maximum and mean channels are almost the same channels. We can conduct these two methods into one activation-to-weight smooth.

LLM.int8() finds that keeping the activation channels with outliers unquantized will help the model maintain its performance. It's interesting that AWQ also finds that keeping that 1% salient channels for weight would protect the performance. We can follow their conclusions for weight-activation quantization.

In our smooth strategy, we calculate the smooth scale $\mathbf{k}_j$ as:

$$\mathbf{z}_j = \mathbf{max}(|\mathbf{X}_{[j,:]}|), \ \mathbf{k}_j = \mathbf{clamp}(\mathbf{z}_j/\mathbf{max}_{0.5\%}(\mathbf{z}), \text{low} = 1) \quad (7)$$

Here, we use the top 0.5% of the largest values as the clipping threshold. We constrain outliers larger than 0.5% by setting their value to 0.5% and employ the resulting scale as an amplifier for quantization-sensitive channels.

## 4 EXPERIMENTS

### 4.1 EXPERIMENTS SETUPS

**Baseline.** We compare with baselines in the A8W4 post-training quantization setting, Zero-Quantv2 (Yao et al., 2023), SmoothQuant (Xiao et al., 2023), RPTQ (Yuan et al., 2023) and ZeroQuant-FP (Wu et al., 2023). Since the quantization schemes vary from different quantization methods, we test one aligned quantization scheme per-token dynamic activation quantization and group-wise weight quantization. Additionally, we include results for static activation quantization to facilitate further comparisons. AWQ (Lin et al., 2023) and GPTQ (Frantar et al., 2022), which is designed for weight-only quantization. As activation quantization is harder for activation, comparison to those methods in A8W4 quantization schemes is unfair. We try to compare the results of A8W4 to A16W3, since they are both the next step of A16W4.

**Models and datasets.** We choose OPT (Zhang et al., 2022) and LLaMA (Touvron et al., 2023a;b) families to evaluate our quantization methods. As BLOOM (Scao et al., 2022) has a similar structure as OPT and have close quantization performance. For Common Sense Question Answers evaluation, we use five zero-shot evaluation task: HellaSwag (Zellers et al., 2019), PIQA (Bisk et al., 2020), Winogrande (Sakaguchi et al., 2021), BoolQ (Clark et al., 2019) and ARC (Clark et al., 2018). Common Sense Question Answers benchmark is done with lm-eval (Lin & Chen, 2023). Follows the evaluation proposed at GPTQ (Frantar et al., 2022), we use WikiText-2 (Merity et al., 2016), PTB (Marcus et al., 1994) and C4 (Raffel et al., 2020) to compare the generation ability.

**Implementation.** We implement our methods with Pytorch Huggingface for the proof of concept. We use the CUTLASS GEMM kernels to develop our two-grained quantization kernels, and we use the INT8 BMM kernels from torch-int.

### 4.2 ACCURACY EVALUATION

**Results on LLaMA Family.** Our static activation quantization surpasses INT3 weight-only quantization, demonstrating that A8W4 quantization offers more sophisticated deployment options than A16W3. Furthermore, our quantized models consistently outperform their half-size counterparts, demonstrating the practical feasibility of dual-grained quantization (DGQ). In addition, our quantization methods consistently produce results that are comparable to, or even superior to, those achieved by ZeroQuantFP, a method that employs FP8 for quantization. This advantage is especially notable when applied to smaller models with identical quantization settings. It's important to note that the utilization of FP8, as reported by van Baalen et al. (2023), comes at the cost of increased chip area and extended inference times. In the realm of Common Sense Question Answering tasks, our dynamic quantization methods outperform other quantization approaches. Additionally, our static quantization schemes yield results that are on par with gradient-based methods (Liu et al., 2023).

**Table 2. Quantization Results on WikiText-2 with A16W3 and A8W4 LLaMA Models**. C4 perplexity results can be found in Table A1 in Appendix A3. † indicates static quantization for activation.

| LLaMA PPL ↓ | | 1-7B | 1-13B | 1-30B | 1-65B | 2-7B | 2-13B | 2-70B |
|---|---|---|---|---|---|---|---|---|
| FP16 | - | 5.68 | 5.09 | 4.10 | 3.53 | 5.47 | 4.88 | 3.31 |
| W3A16 g128 | RTN | 7.01 | 5.88 | 4.87 | 4.24 | 6.66 | 5.51 | 3.97 |
| | GPTQ | 6.55 | 5.62 | 4.80 | 4.17 | 6.29 | 5.42 | 3.85 |
| | AWQ | 6.46 | 5.51 | 4.63 | 3.99 | 6.24 | 5.32 | - |
| W4A8 g128 | RTN | 8.72 | 7.81 | 6.76 | 6.16 | 12.85 | 44.82 | 8.70 |
| | ZeroQuantv2 | 6.44 | 5.32 | 4.36 | - | - | - | - |
| | SmoothQuant | 6.04 | 5.36 | 4.48 | 3.98 | 5.97 | 5.23 | 3.65 |
| | ZeroQuant-FP | 6.32 | 5.26 | **4.26** | - | - | - | - |
| | Ours | **5.85** | **5.21** | 4.28 | **3.71** | **5.64** | **5.01** | **3.45** |
| | Ours† | **6.04** | **5.39** | **4.45** | **3.89** | **5.87** | **5.23** | **3.74** |

**Table 3. CSQA Reuslts on Six zero-shot tasks with A8W4 LLaMA Models.** Due to the unavailability of the identical model as LLM-QAT, we present FP16 accuracy data sourced from LLM-QAT. MMLU results can be found in Table A4 in Appendix A3. † indicates static quantization for activation.

| LLaMA / Acc↑ | Method | PIQA | ARC-e | Arc-c | BoolQ | HellaSwag | Winogrande | Avg. | Δ |
|---|---|---|---|---|---|---|---|---|---|
| LLaMA-1-7B | FP16 | 79.3 | 73.0 | 48.0 | 76.8 | 76.1 | 70.0 | 70.5 | 0.0 |
| | SmoothQunat | 76.0 | 67.4 | 42.8 | 71.0 | 67.8 | 66.0 | 65.2 | 5.3 |
| | LLM-QAT | 77.5 | 70.2 | 45.6 | 74.6 | 73.5 | 67.7 | 68.2 | 2.3 |
| | FP16 | 79.2 | 73.0 | 44.7 | 75.1 | 76.2 | 70.0 | 69.7 | 0.0 |
| | Ours | 78.8 | 72.4 | 43.9 | 74.7 | 74.9 | 70.2 | 69.2 | **0.5** |
| | Ours† | 77.4 | 70.4 | 42.7 | 69.1 | 73.0 | 68.6 | 66.9 | 2.8 |
| LLaMA-1-13B | FP16 | 80.0 | 74.5 | 52.6 | 78.1 | 79.2 | 73.6 | 73.0 | 0.0 |
| | SmoothQunat | 77.1 | 67.4 | 43.4 | 72.5 | 74.3 | 69.5 | 67.4 | 5.6 |
| | LLM-QAT | 79.1 | 73.0 | 51.9 | 77.5 | 77.5 | 70.6 | 71.6 | 1.4 |
| | FP16 | 80.3 | 74.6 | 47.7 | 77.9 | 79.1 | 72.4 | 72.0 | 0.0 |
| | Ours | 80.1 | 73.7 | 46.8 | 77.8 | 78.6 | 71.7 | 71.5 | **0.5** |
| | Ours† | 79.2 | 72.5 | 47.2 | 73.9 | 77.3 | 70.6 | 70.1 | 1.9 |
| LLaMA-1-30B | FP16 | 82.1 | 80.0 | 58.0 | 83.2 | 82.9 | 75.6 | 77.0 | 0.0 |
| | SmoothQunat | 79.5 | 76.5 | 54.5 | 74.9 | 76.9 | 70.6 | 72.2 | 4.8 |
| | LLM-QAT | 80.9 | 80.3 | 56.5 | 81.3 | 81.3 | 76.3 | 76.1 | 0.9 |
| | FP16 | 82.1 | 79.0 | 53.1 | 82.7 | 82.6 | 75.6 | 75.8 | 0.0 |
| | Ours | 81.6 | 78.5 | 53.6 | 83.1 | 82.1 | 74.6 | 75.6 | **0.2** |
| | Ours† | 79.7 | 78.0 | 51.8 | 80.2 | 79.8 | 74.0 | 73.9 | 1.9 |

The application of GLU (Dauphin et al., 2017) in LLaMA amplify the outliers by element-wise multiplication, making quantization difficult. While dynamic activation quantization can mitigate this issue, it necessitates computing statistics for each token before passing through linear kernels, impacting processing time.

**Results on OPT Family.** In our evaluation of OPT models spanning from 125M to 66B parameters, we observe that the gap between static and dynamic activation quantization is relatively narrow. This phenomenon is attributed to the fact that the outliers in OPT models tend to be smaller compared to LLaMA models. Compared to RPTQ, which is also static activation quantization, our methods achieve superior results. As our coarse is finer than RPTQ. But the RPTQ needs to reorder the input channel at each layer norm, introducing extra inference cost. After all, our method outperforms both other quantization methods on OPT Family, demonstrating the generality of our method to different model families and model sizes.

**Table 4. Quantization Results on WikiText-2 with A16W3 and A8W4 OPT Models.** C4 and PTB perplexity can be found in Table A2 and Table A3 in Appendix A3. † indicates static quantization for activation.

| OPT PPL↓ | | 125M | 1.3B | 2.7B | 6.7B | 13B | 30B | 66B |
|---|---|---|---|---|---|---|---|---|
| FP16 | - | 27.65 | 14.63 | 12.47 | 10.86 | 10.12 | 9.56 | 9.34 |
| W3A16 g128 | RTN | 51.22 | 119.00 | 297.98 | 23.54 | 46.03 | 18.80 | 136.89 |
| | GPTQ | 39.24 | 16.47 | 13.69 | 11.65 | 10.35 | 9.73 | 10.96 |
| | AWQ | 36.74 | 16.32 | 13.58 | 11.41 | 10.68 | 9.85 | 9.60 |
| W4A8 g128 | RTN | 32.21 | 17.33 | 15.51 | 51.57 | 3978.101 | 2407.99 | 2832.57 |
| | ZeroQuantv2 | 31.69 | 15.53 | 13.02 | 11.29 | 10.43 | 9.86 | 9.62 |
| | SmoothQuant | **29.01** | **14.71** | 12.71 | **10.90** | **10.25** | 9.57 | 9.32 |
| | RPTQ | - | 15.39 | - | 11.21 | 10.90 | 10.22 | 9.46 |
| | ZeroQuant-FP | - | 15.32 | - | 10.89 | 10.16 | 9.52 | - |
| | Ours | 29.25 | 14.78 | **12.67** | 10.93 | 10.29 | **9.53** | **9.31** |
| | Ours† | **29.94** | **14.96** | **12.75** | **10.92** | **10.30** | 9.55 | 9.32 |

## 4.3 EFFICIENCY EVALUATION

In Figure 4, we compare the end-to-end efficiency of OPT-30B and LLaMa-30B models using different quantization methods: SmoothQuant (A8W8), AWQ (A8W4), and our methods. As SmoothQuant and AWQ both give their implement code, we directly test the implementation time on a single 80G A100 GPU. For shorter sequences, A8W4 implementation takes more time than A8W8 and FP16 due to additional computation introduced by group-wise quantization. However, as sequences get longer, A8W4 outperforms A8W8 because it fuses dequantization and matrix multiplication efficiently. AWQ

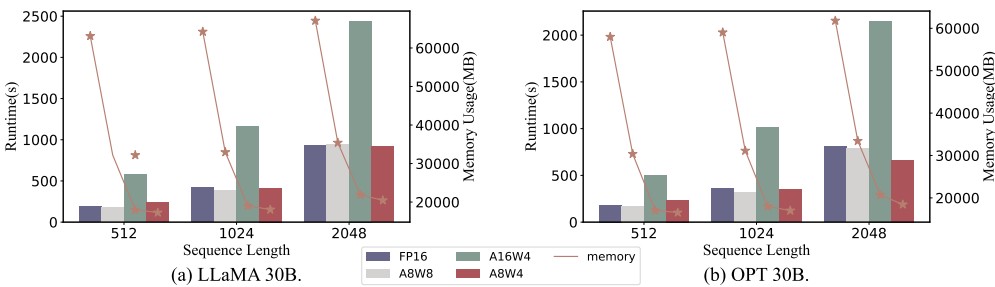

**Figure 4. Runtime and Memory Usage for LLaMA-30B and OPT-30B Across Varying Sequence Lengths on a Single A100-80G GPU.**

(A16W4) performs well but struggles with long sequences due to activation bottlenecks. Our method achieves A8W8-level inference times with half the memory usage, demonstrating its efficiency across various quantization methods.

### 4.4 ABLATION STUDY

**Different quantization scheme for A8W4 LLaMA models.** In our experiments with LLaMA1-7b and LLaMA1-13b models, we explored different quantization schemes, including A8W4 quantization, as detailed in Table 5. The results indicate that fine-grained and coarse-grained quantization methods can lead to up to a 1 PPL (Perplexity) difference. Specifically, in fine-grained quantization, we observe that static activation quantization in combination with group-wise weight quantization outperforms dynamic activation quantization coupled with channel-wise weight quantization. Additionally, our dual-grained quantization approach demonstrates that it introduces minimal additional quantization errors compared to group-wise quantization.

**Table 5. Comparison for different quantization schemes for A8W4 LLaMA models.** S means static tensor-wise quantization, D means dynamic token-wise quantization, CW means channel-wise quantization, GW means Group-wise quantization and DG means Dula-Grained quantization.

| PPL ↓ | LLaMA1-7B | | | | | LLaMA1-13B | | | | |
|---|---|---|---|---|---|---|---|---|---|---|
| | FP16 | S+CW | S+GW | D+CW | **S+DG** | FP16 | S+CW | S+GW | D+CW | **S+DG** |
| WikiText-2 | 5.68 | 6.57 | 6.03 | 6.37 | 6.04 | 5.09 | 6.17 | 5.39 | 5.82 | 5.39 |
| C4 | 7.08 | 8.10 | 7.44 | 7.75 | 7.43 | 6.61 | 7.84 | 6.92 | 7.37 | 6.93 |

**Effect of Percentile Clipping Smooth.** In our analysis, we present the WikiText-2 perplexity (PPL) results in both Table 2 and Table 4. The primary distinction between SmoothQuant (Xiao et al., 2023) and our methods lies in the choice of the smoothing scale. We want to emphasize the effectiveness of our approach, termed Percentile Clipping Smoothing, as a straightforward yet powerful technique for LLaMA quantization. Notably, it's worth mentioning that the presence of outliers in OPT models is less pronounced compared to LLaMA models. This difference explains why our methods achieve comparable performance to SmoothQuant, primarily on OPT models.

### 5 CONCLUSION

In this work, we propose Dual Grained Quantization (DGQ), a promising and hardware-efficient scheme for mixed bit weight-activation quantization (A8W4) for LLM. DGQ is designed to compensate for the hardware-inefficient group-wise quantization by a fine-grained integer quantization scale and a coarse-grained full-precision scale. We also improve the search algorithm for quantization scales to adapt to our quantization scheme. We propose a percentile clipping smooth strategy, achieving a better smooth scales without search. Furthermore, we implement efficient kernels, achieving $3 \times$ speedup over A16W4 for long sequence inference and $0.5 \times$ memory usage over A8W8 with comparable run time.

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

## A1 LIMITATION AND DISCUSSION

In this paper, we propose a simple yet highly efficient quantization method for fine-grained weight-activation quantization. This method makes it practical to implement large language models using A8W4 quantization. Our primary focus is on developing efficient solutions for fine-grained weight quantization. For activation, layer-wise activation quantization strategy (Li et al., 2023) would be a good solution, as the limitation of smooth scales.

In our work, we specifically develop efficient kernels tailored for long-sequence inference. A16W4 kernels will introduce redundant dequantization operations. Self-decoding task is memory bound, A16W4 kernels can ease memory bound. Compared to A16W4, our method have same bit width for weight and the calculation progression is done with INT8 kernels. This theoretically positions our approach as more efficient than A16W4. One challenge we are currently addressing is the need for two separate operations in different situations, and we are actively working on a solution.

Furthermore, self-decoding tasks face a challenge with memory consumption, especially when dealing with large self-attention matrices. For example, with a sequence length of 32K, a single self-attention matrix can occupy about 20GB of memory in FP16. To mitigate this issue, we are exploring quantization methods for activation, which we plan to incorporate into our future work.

## A2 PROOF FOR CONSTRAINTS RELAXATION AND FUSION FOR GRID SEARCH.

The constraints are as follows:

$$\mathbf{W}_{u4} \in [\mathbf{0}, \mathbf{15}], \mathbf{S^{(2)}} \in [\mathbf{0}, \mathbf{15}], \mathbf{W}_{s8} \in [-\mathbf{127}, \mathbf{127}] \tag{A8}$$

The dequantization calculation is as follows:

$$\mathbf{W}_{f16} = (\mathbf{W}_{u4} - \mathbf{ZP}_{u4})\mathbf{S}_{f16}, \mathbf{S}_{f16} = \left\lfloor \mathbf{S}'/\mathbf{S^{(1)}} \right\rceil S^{(1)} \tag{A9}$$

Due to the rounding operation $\lfloor \cdot \rceil$, $\mathbf{S}_{f16}$ may be greater than $\mathbf{S}'$, resulting in maximum values of some groups becoming less than 15. During the search space exploration, the range of $\mathbf{S^{(2)}}$ can exceed 15, and INT4 is unsupported on certain devices. Therefore, we relax the range constraints of $\mathbf{S^{(2)}}$, focusing solely on constraints for $\mathbf{W}_{s8}$ and $\mathbf{W}_{u4}$.

$$\mathbf{W}_{u4} = \left\lfloor \frac{\mathbf{W}_{s8}}{\mathbf{S^{(2)}}} \right\rceil + \mathbf{ZP} \in [\left\lfloor \frac{-\mathbf{127}}{\mathbf{S^{(2)}}} \right\rceil + \mathbf{ZP}, \left\lfloor \frac{\mathbf{127}}{\mathbf{S^{(2)}}} \right\rceil + \mathbf{ZP}] \tag{A10}$$

We can merge the two intervals as follows:

$$\mathbf{W}_{u4} \in [\mathbf{max}(\mathbf{0}, \lfloor \frac{-\mathbf{127}}{\mathbf{S^{(2)}}} \rceil + \mathbf{ZP}), \mathbf{min}(\mathbf{15}, \lfloor \frac{\mathbf{127}}{\mathbf{S^{(2)}}} \rceil + \mathbf{ZP})] \tag{A11}$$

## A3 MORE ACCURACY RESULTS

In this section, we provide a comprehensive presentation of our results across various datasets to complement the main paper. We also provide here a comparison with contemporaneous work Omniquant (Shao et al., 2023) and FPTQ (Li et al., 2023). Specifically, the results include:

- C4 perplexity in the LLaMA families (Table A1)
- C4 perplexity in OPT families (Table A2).
- PTB perplexity in OPT families (Table A3).
- MMLU in LLaMA families (Table A4).

**Table A1. Quantization Results on c4 with A16W3 and A8W4 LLaMA Models**. † indicates static quantization for activation.

| LLaMA PPL ↓ | | 1-7B | 1-13B | 1-30B | 1-65B | 2-7B | 2-13B | 2-70B |
|---|---|---|---|---|---|---|---|---|
| FP16 | - | 7.08 | 6.61 | 5.98 | 5.62 | 6.97 | 6.46 | 5.52 |
| W3A16 g128 | RTN | 8.62 | 7.49 | 6.58 | 6.10 | 8.40 | 7.18 | 6.02 |
| | GPTQ | 7.85 | 7.10 | 6.47 | 6.00 | 7.89 | 7.00 | 5.85 |
| | AWQ | 7.92 | 7.07 | 6.37 | 5.94 | 7.84 | 6.94 | - |
| | OmniQuant | 7.34 | 6.76 | 6.11 | 5.73 | 7.35 | 6.65 | 5.86 |
| W4A8 g128 | RTN | 10.76 | 9.94 | 8.14 | 7.96 | 17.29 | 90.57 | 11.86 |
| | ZeroQuantv2 | 7.79 | 6.78 | 6.16 | - | - | - | - |
| | SmoothQuant | 7.51 | 6.89 | 6.39 | 5.94 | 7.50 | 6.82 | 5.78 |
| | ZeroQuant-FP | 7.51 | 5.73 | **6.09** | - | - | - | - |
| | Ours | **7.29** | **6.73** | 6.10 | **5.73** | **7.16** | **6.62** | **5.62** |
| | Ours† | **7.43** | **6.93** | **6.31** | **5.97** | **7.44** | **6.82** | **5.89** |

**Table A2. Quantization Results on c4 with A16W3 and A8W4 OPT Models** † indicates static quantization for activation.

| OPT PPL ↓ | | 125M | 1.3B | 2.7B | 6.7B | 13B | 30B | 66B |
|---|---|---|---|---|---|---|---|---|
| FP16 | - | 24.61 | 14.73 | 13.17 | 11.75 | 11.21 | 10.69 | 10.28 |
| W3A16 g128 | RTN | 40.13 | 126.47 | 372.23 | 32.56 | 44.12 | 25.70 | 286.87 |
| | GPTQ | 30.08 | 16.47 | 14.54 | 12.48 | 11.58 | 10.91 | 11.35 |
| | AWQ | 30.39 | 16.27 | 14.19 | 12.30 | 11.61 | 10.96 | 10.53 |
| | OmniQuant | 29.34 | 16.11 | 14.15 | 12.31 | 11.63 | 10.98 | 10.51 |
| W4A8 g128 | RTN | 27.93 | 17.52 | 16.33 | 98.34 | 3926.05 | 3557.30 | 2493.73 |
| | ZeroQuantv2 | 27.19 | 15.73 | 13.82 | 12.19 | 11.64 | 11.00 | 10.63 |
| | SmoothQuant | **25.99** | 15.16 | 13.46 | 11.88 | **11.39** | 10.75 | 10.32 |
| | RPTQ | - | 15.48 | - | 12.11 | 11.62 | 11.01 | 10.57 |
| | ZeroQuant-FP | - | 15.32 | - | 11.95 | 11.30 | 10.75 | - |
| | Ours | 26.03 | **15.10** | **13.42** | **11.87** | 11.40 | **10.74** | **10.33** |
| | Ours† | **26.64** | **15.24** | **13.45** | **11.89** | **11.42** | **10.76** | **10.33** |

**Table A3. Quantization Results on ptb with A16W3 and A8W4 OPT Models** † indicates static quantization for activation.

| OPT PPL ↓ | | 125M | 1.3B | 2.7B | 6.7B | 13B | 30B | 66B |
|---|---|---|---|---|---|---|---|---|
| FP16 | - | 32.55 | 16.97 | 15.11 | 13.09 | 12.34 | 11.84 | 11.36 |
| W3A16 g128 | RTN | 64.67 | 222.13 | 337.75 | 39.90 | 65.33 | 34.27 | 309.69 |
| | GPTQ | 45.17 | 19.90 | 17.06 | 14.24 | 12.84 | 12.54 | 13.27 |
| | AWQ | 44.07 | 19.59 | 16.52 | 13.98 | 12.87 | 66.68 | 3.4e3 |
| | OmniQuant | 45.29 | 20.42 | 17.08 | 14.23 | 13.49 | 12.54 | 12.06 |
| W4A8 g128 | RTN | 38.31 | 20.84 | 19.75 | 65.86 | 3370.84 | 2972.69 | 2556.84 |
| | ZeroQuantv2 | 36.66 | 18.35 | 16.11 | 13.70 | 12.91 | 12.28 | 11.84 |
| | SmoothQuant | 34.32 | **17.37** | **15.27** | 13.27 | **12.55** | 11.93 | 11.42 |
| | RPTQ | - | 17.79 | - | 13.74 | 13.40 | 12.41 | 11.73 |
| | ZeroQuant-FP | - | 18.19 | - | 13.44 | 12.55 | 11.90 | - |
| | Ours | **34.29** | 17.48 | 15.31 | **13.26** | 12.61 | **11.93** | **11.42** |
| | Ours† | **35.29** | **17.61** | **15.34** | **13.29** | **12.63** | **11.93** | **11.42** |

**Table A4. MMLU Reuslts with A8W4 LLaMA Models.** † indicates static quantization for activation.

| LLaMA / Acc↑ | Method | BW | Humans | STEM | Social | Other | **Avg.** | Δ |
|---|---|---|---|---|---|---|---|---|
| | FP16 | A16W16 | 33.60 | 31.10 | 38.20 | 38.40 | 35.20 | 0.00 |
| | SmoothQunat | A8W8 | 33.80 | 30.32 | 37.63 | 39.08 | 35.14 | 0.06 |
| | GPTQ | A16W4 | 32.39 | 30.35 | 35.03 | 36.15 | 33.40 | 1.80 |
| LLaMA-1-7B | FPTQ | A8W4 | 30.20 | 29.95 | 32.76 | 35.87 | 32.02 | 3.18 |
| | FP16 | A16W16 | 31.27 | 30.53 | 36.79 | 35.90 | 33.50 | 0.00 |
| | Ours | A8W4 | 31.08 | 30.53 | 37.09 | 34.56 | 33.78 | **-0.28** |
| | Ours† | A8W4 | 28.96 | 30.22 | 35.91 | 34.24 | 32.20 | 1.30 |
| | FP16 | A16W16 | 44.60 | 37.10 | 54.00 | 53.50 | 47.10 | 0.00 |
| | SmoothQunat | A8W8 | 44.14 | 36.51 | 54.05 | 52.65 | 46.64 | 0.54 |
| | GPTQ | W4A16 | 46.01 | 39.00 | 54.01 | 53.36 | 47.96 | -0.86 |
| LLaMA-1-13B | FPTQ | W4A8 | 40.96 | 34.19 | 49.72 | 49.75 | 43.46 | 3.64 |
| | FP16 | A16W16 | 40.73 | 38.31 | 54.60 | 54.04 | 47.06 | 0.00 |
| | Ours | A8W4 | 40.93 | 37.69 | 50.44 | 51.48 | 45.83 | 1.23 |
| | Ours† | A8W4 | 39.56 | 41.50 | 48.66 | 51.27 | 45.74 | 1.32 |
| | FP16 | A16W16 | 61.80 | 52.00 | 73.30 | 67.60 | 63.50 | 0.00 |
| | SmoothQunat | A8W8 | 61.32 | 50.50 | 71.69 | 66.90 | 62.56 | 0.94 |
| | GPTQ | A16W4 | 60.23 | 52.09 | 72.15 | 66.75 | 62.60 | 0.90 |
| LLaMA-1-65B | FPTQ | A8W4 | 59.85 | 49.24 | 71.50 | 65.89 | 61.52 | 1.98 |
| | FP16 | A16W16 | 57.72 | 47.04 | 75.96 | 67.44 | 62.18 | 0.00 |
| | Ours | A8W4 | 56.76 | 43.62 | 75.07 | 65.62 | 61.21 | **0.97** |
| | Ours† | A8W4 | 55.60 | 44.86 | 72.11 | 63.53 | 59.57 | 2.61 |

