# OpenReview forum: "Dual Grained Quantization: Efficient Fine-grained Quantization for LLM"
_ICLR.cc/2024/Conference — ICLR 2024 Conference Withdrawn Submission_

### Official Review · Reviewer_h9HQ · 2023-10-24

**Soundness:** 2 fair
**Presentation:** 2 fair
**Contribution:** 1 poor
**Rating:** 3
**Confidence:** 5

**Summary:**

This paper introduces Dual Grained Quantization (DGQ), a novel A8W4 quantization method for Large Language Models (LLMs), addressing the trade-off between quantization loss and inference efficiency. DGQ combines fine-grained INT4 weight quantization with coarse-grained INT8 matrix multiplication, supported by a two-phase grid search algorithm and a percentile clipping schema. Experimental results show that DGQ consistently outperforms previous methods, achieving significant memory reduction and speed gains, making it a promising approach for efficient deployment of A8W4 LLMs in practical applications.

**Strengths:**

+ The proposed DGQ can accelerate the model on general-purpose hardware and avoid designing the specific hardware.
+ How to effectively quantize the LLMs is important. The main idea in this paper is easy to follow and looks reasonable.
+ The two-stage grid search seems work well in the experiments.

**Weaknesses:**

- The novelty of this paper is poor. The DGQ is incremental from existing quantization methods. It is extremely unclear why the proposed approach is suitable for LLMs as it can be also leveraged to quantize other models. The authors should analyze the LLMs and provide the motivation that the proposed DGQ is unique for LLMs, like how AWG does.
- Fig. 1 looks unfair to other methods. I think if you set other methods as the same A8W4, the memory usage should be similar. Hence, this comparison cannot be the benefit of the proposed DGQ.
- This paper lacks many analysis, all the techniques are presented straightforward. The authors are encouraged to provide the motivation of each component why it does work.
- The two-phase grid search seems time-consuming.
- The performance is not promising in practice. In Table 2, DGQ only outperforms SmoothQunat slightly. In Table 3, DGQ looks inferior to SmoothQunat.
- The quality of the figures is poor. It is not that easy to get the meaning of the symbols and equations.

**Questions:**

Why the proposed DGQ is unique for LLMs?

---

### Official Review · Reviewer_XQ5m · 2023-10-30

**Soundness:** 3 good
**Presentation:** 1 poor
**Contribution:** 2 fair
**Rating:** 3
**Confidence:** 3

**Summary:**

This paper proposes Dual Grained Quantization (DGQ) for LLM quantization. For the quantization format, DGQ adopts a fine-grained (group-wise) 4-bit weight quantization. During the inference, DGQ dequantizes the weights back to INT8 weights with coarser-grained scale groups (channel-wise), so that the computation can be efficiently executed. The weights have fine-grained INT8 scales and coarse-grained FP16 scales, and DGQ uses a 2-stage grid search to determine them. Experiments are conducted on OPT and LLaMA-v1/v2 models. This paper also shows that the proposed W4A8 quantization scheme with kernel design can achieve a 3x speedup than the W4A16 baseline.

**Strengths:**

* W4A8 quantization solution for LLM inference is a practical choice. The overall method is practical in my opinion.
* The paper experiments with a kernel implementation.
* The paper compares with both weight-only and weight-activation quantization baselines.

**Weaknesses:**

My major concern is the current paper writing is hard to follow, with vague statements and logic. This causes me to have many doubts after reading the paper, see the questions section.

Moreover, as the major contribution of this paper is the dual-grained format design more friendly for kernel implementation, providing the kernel implementation as well as showing detailed GPU profiling can be helpful.

**Questions:**

* The motivation for using the "dual-grained format" needs more justifications. The introduction said that "Partitioning the integer GEMM operation into discrete segments and aggregating them through floating-point arithmetic is inefficient on existing hardware infrastructure". I recommend justifying the claim with detailed kernel analyses.
* Section 3.4 reviews the prior literature on addressing the activation outliers. However, it's not clear why this work chooses to use a quantile to decide the scale, and why is the "percentage clipping smoothing" better than others?
* Many notations are not well-defined, for example, in Section 3.4.
* In Equation 2, an immediate question is how to handle the overflow when quantizing the multiplication of uINT4 and INT8 to INT8. The question is not mentioned and answered until Eq. (5), I think it should be referred to earlier.
* Is the reported acceleration ratio only for the prefill stage or the entire inference? I recommend the authors analyze the acceleration ratios for the prefill stage and decoding stage independently.
* The results section says that "as sequences get longer, A8W4 outperforms A8W8 because it fuses dequantization and matrix multiplication efficiently". Smoothquant does not conduct the dequantization process and conduct per-tensor quantization, why can this method become faster than Smoothquant A8W8 as the sequence length goes up?
* It is recommend to compare the method with other SoTA work, such as
  -	Yuan, Zhihang, et al. "RPTQ: Reorder-based Post-training Quantization for Large Language Models." arXiv preprint arXiv:2304.01089 (2023).
  -	Wei, Xiuying, et al. "Outlier Suppression+: Accurate quantization of large language models by equivalent and optimal shifting and scaling." arXiv preprint arXiv:2304.09145 (2023).
  -	Shao, Wenqi, et al. "Omniquant: Omnidirectionally calibrated quantization for large language models." arXiv preprint arXiv:2308.13137 (2023).


Minor:
* Some grammar mistakes, e.g., Figure 1 caption, "our A8W4 implement matain the comparable ..."

---

### Official Review · Reviewer_FXW6 · 2023-10-31

**Soundness:** 2 fair
**Presentation:** 1 poor
**Contribution:** 1 poor
**Rating:** 3
**Confidence:** 4

**Summary:**

This paper introduces the W4A8 quantization/acceleration approach, which integrates weight-only quantization (e.g., W4A16) with INT8 quantization (e.g., W8A8). By employing the INT8 kernel for LLMs, we enhance the throughput and latency during LLM inference. Conversely, using the dequantization kernel for W4A16 formats optimizes the memory-bound workload of decoder-only transformers. The goal of merging these techniques is to streamline inference for LLMs, such as OPT or LLaMa. Given that dual quantization is applied to weight parameters, this study also incorporates a grid search algorithm. Ultimately, their findings highlight improved runtime speeds and maintain model performance post-compression.

**Strengths:**

- This paper combines two types of famous and current quantization approaches.
- This paper shows comparable and extensive results on LLaMa and OPT models.

**Weaknesses:**

1) Concerns Regarding Acceleration Results:

The foundation of the presented method appears to lie in its kernel design and the W4A8 dequantization results. Typically, quantization techniques are employed to enhance latency and throughput. Yet, there are instances where quantization may not favorably influence acceleration outcomes. For instance, LLM.int8() doesn't seem to offer significantly better kernel results based on its decomposition method. When examining larger batch sizes, both LUT-GEMM and AWQ kernels don't seem to accelerate the generation steps. With this in mind, the paper should provide a comprehensive evaluation of acceleration and kernel design.

The paper lacks details concerning 'runtime'. There's an absence of discussion on both summarization and generation steps. While the INT8 kernel can potentially enhance the summarization and generation steps, the weight-only quantization kernel appears to improve only the generation step. Unfortunately, the paper doesn't delve into these issues.

In the "SmoothQuant" paper, it's stated that latency doesn't accelerate beyond 1.5x. How, then, does this paper claim a 3.24x speed gain? Given that this paper's kernel is implemented within the Huggingface Framework, what assurances do we have that these results are solely attributed to the custom kernel?

Considering larger batch sizes—where the INT8 kernel is presumed to be particularly effective—how does this method maintain the efficacy of the dequantization kernel (like AWQ or LUT-GEMM)?

2) Concerns on Model Performance Post-Quantization:

Over the past year, many studies have opted to showcase common sense reasoning or MMLU results as benchmarks for their quantization methods. This is in preference to using the wikitext PPL, especially when highlighting the maintained performance of quantization methods for generative AI. Although this paper does present CSQA results in Table 3, it's noticeable that there's no mention of results from large models or LLaMa-2 models. Furthermore, MMLU results have been relegated to the appendix. Why hasn't the paper included thorough experimentation on LLaMa-2 and larger models using the CSQA or MMLU dataset?

**Questions:**

included in weaknesses.

---

### Official Review · Reviewer_s9uD · 2023-11-07

**Soundness:** 2 fair
**Presentation:** 2 fair
**Contribution:** 2 fair
**Rating:** 5
**Confidence:** 4

**Summary:**

This paper introduces Dual Grained Quantization (DGQ) for Large Language Models (LLMs) to address the hardware challenges in memory and computational efficiency. DGQ quantizes the model weights into 4-bit integers for better memory efficiency, with two-level scaling factors: one fined-grained UINT4 scale factor and one coarse-grained FP16 scale factor. During computation, DGQ dequantizes the weight into INT8 and performs GEMM using INT8 kernels for better computational efficiency. DGQ also applies a two-phase grid search to optimize the quantization range and proposes a percentile clipping for smoothing the activation outliers. Experiments show that DGQ achieves 1.12x memory reduction and 3.24x speed gains compared to 4-bit weight-only quantization with similar accuracy and perplexity.

**Strengths:**

+ The paper is well-organized and easy to follow.
+ The proposed percentile clipping smoothing is very interesting. It combines both clipping and smoothing into one smooth scale.
+ The results of W4A8 with per-token activation quantization without group quantization are impressive.
+ Evaluation experiments on real GPU machines with well-implemented kernels look very solid, and the measured runtime and memory usage are very promising.

**Weaknesses:**

+ The paper writing is not clear enough. Many details in the proposed techniques are missing, such as the granularity of activation quantization, and the calibration dataset. Please see the questions below.
+ The novelty of dual-level quantization is limited. Using two-level scaling factors (UINT4 for group scaling and FP for channel scaling) in quantization was first proposed in VSQuant and has been used in many other works, including QLoRA.
+ The novelty of the proposed two-phase search for the quantization range alpha is also limited. Grid searching for quantization range alpha has been used in AWQ.
+ The evaluation section lacks a detailed ablation study on different techniques used in DGQ, such as improvement breakdown on static/dynamic quantization, group-wise/dual-grained quantization, AbsMax smooth/percentile clipping smooth.

**Questions:**

+ What is the granularity for dynamic activation quantization?
+ In Table 5, what is the performance of D+GW, D+DG?
+ How much improvement does percentile clipping smoothing bring to DGQ?
+ What is time cost for the two-phase grid search for quantization range alpha?
+ What is the calibration setting and evaluation setting for the experiments?
+ Why W4A8 RTN results are so bad for LLaMa 2 in Table 1 and Table 2?